# Redefining the Taxonomic Boundaries of Genus *Xanthomonas*

Kanika Bansal [†] , Sanjeet Kumar [†], Anu Singh [†], Arushi Chaudhary and Prabhu B. Patil *

Bacterial Genomics and Evolution Laboratory, CSIR-Institute of Microbial Technology, Chandigarh 160036, India;
ksanjeet.ibab@gmail.com (S.K.)
* Correspondence: pbpatil@imtech.res.in
† These authors contributed equally to this work.

**Abstract:** The genus *Xanthomonas* primarily comprises phytopathogenic species. By carrying out deep phylo-taxonogenomics, we recently reported that the genera *Xylella*, *Stenotrophomonas*, and *Pseudoxanthomonas* are misclassified and belong to the genus *Xanthomonas*. Considering the importance of *Xanthomonas*/*Xylella* as plant pathogens and to further determine the taxonomic and phylogenetic breadth of this genus, we extended our earlier study by including all the reported genera and families in the order. This investigation revealed that at least four more genera belong to the genus *Xanthomonas*, with a notable case being *Lysobacter*, after which the family and order are named. Similarly, our investigation also allowed us to reveal the expanded taxonomic breadth of the related genus *Rhodanobacter*. This finding of a major related genus that lacks plant pathogenic species will allow for taxonomy-based comparative studies. The phylo-taxonogenomic revelations were further supported by complete 16S rRNA-based sequence boundaries proposed for genus delineation. Accordingly, we propose a taxonomic revision of these major and closely related genera along with their constituent families within the order *Lysobacteraceae* (*Xanthomonadaceae*). The identification of a major related genus lacking plant pathogenic species will be important in investigating the origin and success of pathogenic species/lineages in the genus *Xanthomonas*.

**Keywords:** *Xanthomonas*; *Rhodanobacter*; *Lysobacteraceae*; *Lysobacterales*; *Frateuriaceae*; taxonogenomics; phylogenomics

## 1. Introduction

According to the current standing in nomenclature, the order *Lysobacterales* (also known as *Xanthomonadales*) constitutes two families, namely, *Lysobacteraceae* and *Rhodanobacteraceae*. Both of these complex families consist of more than 30 genera and around 300 species [1]. These include diverse species of phytopathogens, environmental bacteria, and opportunistic human pathogens [2]. The *Lysobacteraceae* (*Xanthomonadaceae*) family contains *Xanthomonas* and *Xylella*, the two major phytopathogenic genera with significant economic and agricultural impacts [3–6]. The member species of the *Xanthomonas* and *Xylella* genera infect a wide range of crops and plants across the globe. On the other hand, the family also includes the metabolically versatile genus *Stenotrophomonas*, with one of its species, *S. maltophilia*, being a multidrug-resistant opportunistic pathogen responsible for hospital-acquired infections in immunocompromised patients [7]. Interestingly, *Stenotrophomonas* was earlier classified as *Xanthomonas*, and there are an increasing number of reports of *Stenotrophomonas* spp. as plant pathogens [8–10]. At the same time, there are several reports of novel species of *Xanthomonas* that are non-pathogenic in nature [8,9], suggesting plant associated and environmental lifestyle of *Xanthomonas*. Other environmental genera in this family include *Pseudomarimonas*, *Lysobacter*, and *Vulcaniibacterium*, which consist of species of industrial importance causing the leaching of heavy metals and bioremediation [10–12]. The synthesis of extracellular enzymes by members of the *Lysobacter* group has attracted considerable attention. The group is also considered as a prolific source for the creation of novel antibiotics, including β-lactams [13], macrocyclic lactams,

and antibiotics that contain macrocyclic peptides or depsipeptides, such as katanosins [14]. Recent research has shown that *Lysobacter* species, including *L. enzymogenes* C3, have the potential to act as biological pest controllers for plant diseases [15]. On the other hand, the family *Rhodanobacteraceae* [1] consists of genera encoding functions required for the bioremediation of hydrocarbons, etc. [16,17]. Members of the genus *Dyella* act as potential biocontrol agents against grapevine yellows [18]. Genera like *Fulvimonas* and *Aerosticca* are isolated from soil after enrichment with acetylated starch plastic and from crude oil-contaminated soil [17]. The species of the genus *Oleiagrimonas*, like *Oleiagrimonas soli*, play a special role in oil contamination bioremediation [19]. The member species of *Rhodanobacter* like *R. denitrificans* and *R. thiooxydans* are acid-tolerant denitrifiers that are well-suited to acidic, nitrate-rich subsurface conditions. pH is proven to be the primary driver of bacterial community structure in this contaminated subterranean environment [20]. In acidic tundra soils that were experimentally modified with long-term nutrient fertilization, Campbell and co-workers reported that bacteria belonging to the *Dyella* and *Rhodanobacter* lineages were present in significant abundance [21].

The integration of classical taxonomy with genomic evidence has revolutionized the field of microbial systematics [22,23]. Apart from species-level reclassifications, the implementation of genome-based methods has enabled reconciliations on the order, family, and genus levels [24,25]. Average nucleotide identity (ANI) and digital DNA-DNA hybridization (dDDH) are widely used for species delineation; however, these indices are not ideal for genus-level taxonomy because of their reliance on nucleotide identity [26,27]. Average amino acid identity (AAI) [28] and the percentage of conserved proteins (POCP) [29] have been proposed for genus-level classification. AAI is based on the similarity calculation of protein sequences common to the two bacterial isolates and allows for the comparison of diversely related isolates. Unlike nucleotide information, the conservation reflected in the amino acid sequence similarities enables the delineation of high taxa, especially genera [24,28]. However, the POCP only accounts for the presence or absence of proteins and is not suitable when the relatives have a reduced genome size, as in the case of *Xylella*, which was reported as a variant *Xanthomonas* lineage with a highly reduced genome size and G+C content [30]. Hence, AAI is more robust for investigating taxonomy on the level of the genus, irrespective of a reduced genome size and G+C content [8,26,28]. Though the genus level threshold for AAI is assigned as 60–80% [26,31,32], Meehan and co-workers used 65% and Zheng et al. used 68% as cut-offs for the genus delineation of the genera *Mycobacterium* and *Lactobacillus*, respectively [24,25]. Furthermore, whereas AAI is typically calculated from all the genes shared between a pair of organisms [28,33], the AAI values for the core genes (cAAI) are employed as a robust parameter for genus delineation. In addition to these genome similarity criteria, whole genome-based phylogeny provides us with a robust phylogenomic framework with which to verify appropriate AAI criteria based on the specific thresholds or boundaries of identified lineage(s) that do not violate the monophyly rule [34–36]. Hence, there is an excellent opportunity and need to understand the taxonomic boundaries of the genus *Xanthomonas* within the family *Lysobacteraceae* (*Xanthomonadaceae*) and the order *Lysobacterales* (*Xanthomonadales*).

The current taxonomy of the order *Lysobacterales* (*Xanthomonadales*) is based on 16S rRNA analysis, a limited number of molecular markers, conserved sequence indels (CSI), and a phylogenetic tree based on no more than 30 conserved proteins [1]. In an earlier study, by carrying out an in-depth phylo-taxonogenomic investigation of the order Lysobacterales, we reported major reshufflings on the family level, in addition to revealing the boundary of the order and its outliers [10]. In a follow-up study, by carrying out an in-depth phylo-taxonogenomic investigation of the genus *Xanthomonas* and its close relatives, we also reported that the member species of the genera *Pseudoxanthomonas*, *Stenotrophomonas*, and *Xylella* belong to a single genus [30]. As an extension of this study, herein, we re-evaluate the genus boundary of *Xanthomonas* by including all the species of all the reported genera within the family and the order *Lysobacterales* (*Xanthomonadales*). In addition to the genus *Xanthomonas* and its family, this study also allowed us to revise the

taxonomic and phylogenetic breadth of the genus *Rhodanobacter* and its family within the order *Lysobacterales*.

## 2. Materials and Methods

### 2.1. Genome Access and Quality Assessment

All the information about the members within the order *Lysobacterales* (*Xanthomonadales*) was obtained from their LPSN (List of Prokaryotic Names with Standing in Nomenclature) (https://lpsn.dsmz.de/order/lysobacterales, accessed on 14 October 2021). The genomes of the strains used in the study were obtained from NCBI microbes (https://www.ncbi.nlm.nih.gov/genome/microbes/, accessed on 14 October 2021). The quality check of the genomes was carried out using CheckM v1.1.3 [37]. The genomes that passed the QC (<5% contamination and >95% completeness) were considered and annotated using Prokka v1.14.6 [38].

### 2.2. Phylogenomic Investigation of the Members of the Order Lysobacterales (Xanthomonadales)

The core genome phylogeny was generated using PhyloPhlAn 3.0, which is based on nearly 400 conserved genes. The PhyloPhlAn tool employs a phylogenetic pipeline that is both modular and parallel and can be customized to suit the user's needs. The pipeline commences with the identification of phylogenetic markers from the input sequences, culminating in the inference of the final tree [39]. Here, USEARCH v5.2.32 [40] was implemented for ortholog searching, MUSCLE v3.8.3 [41] was used for multiple sequence alignment, and FastTree v2.1 [42] was used for phylogenetic construction. In order to obtain a more robust phylogeny, we fetched the core gene using PIRATE (GNU GPL v3.0) [43]. PIRATE is suitable for the identification of ortholog groups of divergent genomes using amino acid identity thresholds of 50%, 60%, 70%, 80%, 90%, and 95%. PIRATE executes a pangenome pipeline to fetch the core genome with a high level of robustness (https://figshare.com/s/bcc8e94bca580fbb57f8).

### 2.3. Taxonogenomic Assessment of the Members of the Order Lysobacterales (Xanthomonadales)

The genome relatedness amongst the genomes was assessed using the average amino acid identity (AAI) with CompareM v0.0.23 (https://github.com/dparks1134/CompareM), which uses the mean amino acid identity of orthologous genes between a given pair of genomes. Furthermore, for the core average amino acid identity (cAAI), the core genes amongst the type/representative species genomes were fetched from the PIRATES pangenome analysis (https://figshare.com/s/bcc8e94bca580fbb57f8). These core genes were then used to evaluate the cAAI values. The 16S rRNA phylogeny and similarity were generated using BLASTn. All the complete 16S rRNA sequences used in this study were obtained from LPSN (https://lpsn.dsmz.de/).

## 3. Results and Discussion

### 3.1. Phylogenomic Evaluation of the Families and Their Boundaries within the Order Lysobacterales (Xanthomonadales)

A total of 213 genomes belonging to type species and type strains of the order and families within *Lysobacterales* (*Xanthomonadales*) are summarized in the table (Supplementary Table S3). We also evaluated all the conserved genes of 213 species using fastANI, validating that there are no synonyms on the species level (https://figshare.com/s/bcc8e94bca580fbb57f8). Species of the genus *Xanthomonas* have a genome size of 5 Mb, whereas *Pseudoxanthomonas* and *Stenotrophomonas* have 3–5 Mb genomes, and *Xylella*'s genome is 2.5 Mb in size. Other members of the order, such as *Lysobacter*, *Luteimonas*, *Thermomonas*, and *Vulcaniibacterium*, have genomes ranging in size from 2 to 4 Mb. The average GC content of species of the genera *Xanthomonas* and *Pseudoxanthomonas* and *Stenotrophomonas* genomes is 60–70%, whereas species of the genus *Xylella* have a reduced GC of around 51%. Other species of this genus have a GC content of 65–71%. In contrast, other species of the remaining genera, such as *Thermomonas*, *Rhodanobacter*, *Frateuria*, *Dyella*, *Luteibacter*, etc., have 2.5–5 Mb genomes with a 60–65% GC.

A whole-genome-based phylogeny obtained using PhyloPhlAn, including the type species of all the genera reported in the order *Lysobacterales* (*Xanthomonadales*) (n = 33), revealed two major clades corresponding to the reported families of *Lysobacteraceae* (*Xanthomonadaceae*) and *Rhodanobacteraceae* (Figure 1). *Pseudofulvimonas gallinarii* and *Ahniella affigens* formed outgroups for these families, respectively. A whole-genome-based phylogeny, including all the type species and type strains (n = 213) of all the reported genera of the order *Lysobacterales* (*Xanthomonadales*), also revealed two major clades corresponding to the families of *Lysobacteraceae* (*Xanthomonadaceae*) and *Rhodanobacteraceae* (Figure 2). PhyloPhlAn uses more than 400 conserved genes from across the bacterial world, thereby providing a robust phylogenomic tree. The phylogenomic positioning of the constituent species was further confirmed using a core pangenome-based tree parameter. The implementation of PIRATE (pangenome investigation) resulted in core genome extraction from all the strains under study (n = 213). A phylogenomic tree obtained using FastTree resulted in a phylogenomic tree similar to that obtained using PhyloPhlAn and PIRATE (Figures 2 and 3). In all three phylogenomic trees' construction, we used *Ignatzshineria larvae* DSM 13226^T and *Pseudomonas aeruginosa* DSM50071^T as the outgroups. In these trees, *Pseudofulvimonas gallinarii* and *Ahniella affigens* also formed outgroups for these families, as mentioned earlier. In the current study, we are not proposing them as distinct families, and at the same time, we are excluding them from the known families based on the phylogenomic trees (Figures 1–3). However, we are including them in the order.

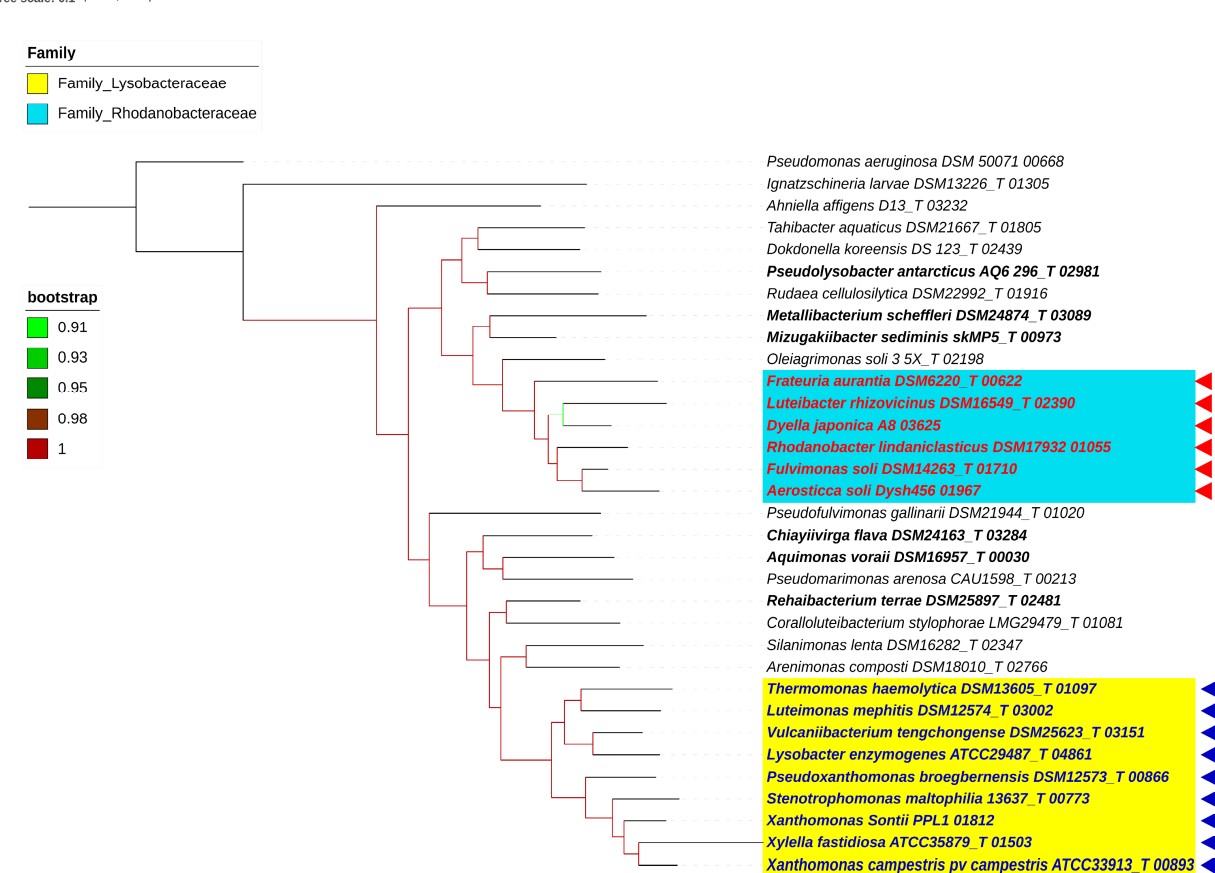

**Figure 1.** PhyloPhlAn of type species. PhyloPhlAn tree comprising 33 type species and two outgroups, *Pseudomonas aeruginosa* DSM 50071 00668^T and *Ignatzschineria larvae* DSM13226^T 01305. Yellow represents the *Lysobacteraceae* family, with blue triangles representing proposed species in the genus *Xanthomonas*. The sky-blue tint represents the *Rhodanobacteracea* family, while the red triangles represent the proposed species of the *Rhodanobacter* genus. The bootstrap values are displayed with color branches. Reshuffled genera are shown in bold.

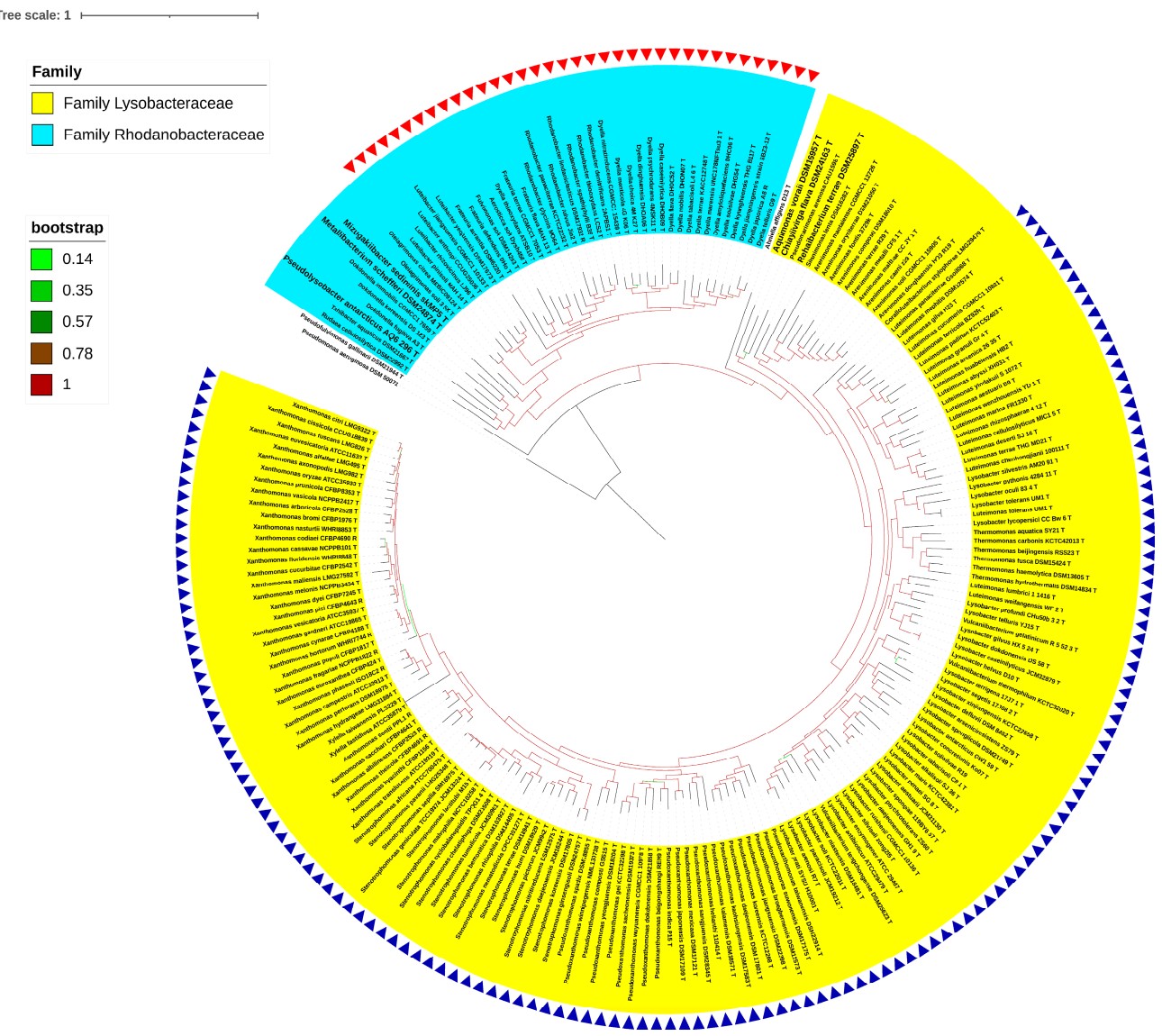

**Figure 2.** PhyloPhlAn of type strains. PhyloPhlAn tree depicting 213 Type strains, with *Pseudomonas aeruginosa* DSM 50071 00668[T] serving as an outgroup. Yellow shade represents the *Lysobacteraceae* family, with blue triangles representing the proposed species belonging to the genus *Xanthomonas*. The *Rhodanobacteraceae* family is represented by the sky-blue hue, while the proposed species belonging to the genus *Rhodanobacter* are represented by the red triangles. The bootstrap values are displayed as numbers with color branches.

## 3.2. Reshuffling of Six Genera within the Families Lysobacteraceae and Rhodanobacteraceae

Based on the previous results, we propose moving the genera *Aquimonas* Saha et al. 2005 [44], *Chiayiivirga* Hsu et al. 2013 [45], and *Rehaibacterium* Yu et al. 2013 [46] from the family *Rhodanobacteraceae* to the family *Lysobacteraceae* and reshuffling *Mizugakiibacter* Kojima et al. 2014 [47], *Metallibacterium* Ziegler et al. 2013 [48], and *Pseudolysobacter* Wei et al. 2020 [49] from the family *Lysobacteraceae* to the family *Rhodanobacteraceae.* These member genera of this family were identified through the deep order-level phylo-taxonogenomic analysis in this study.

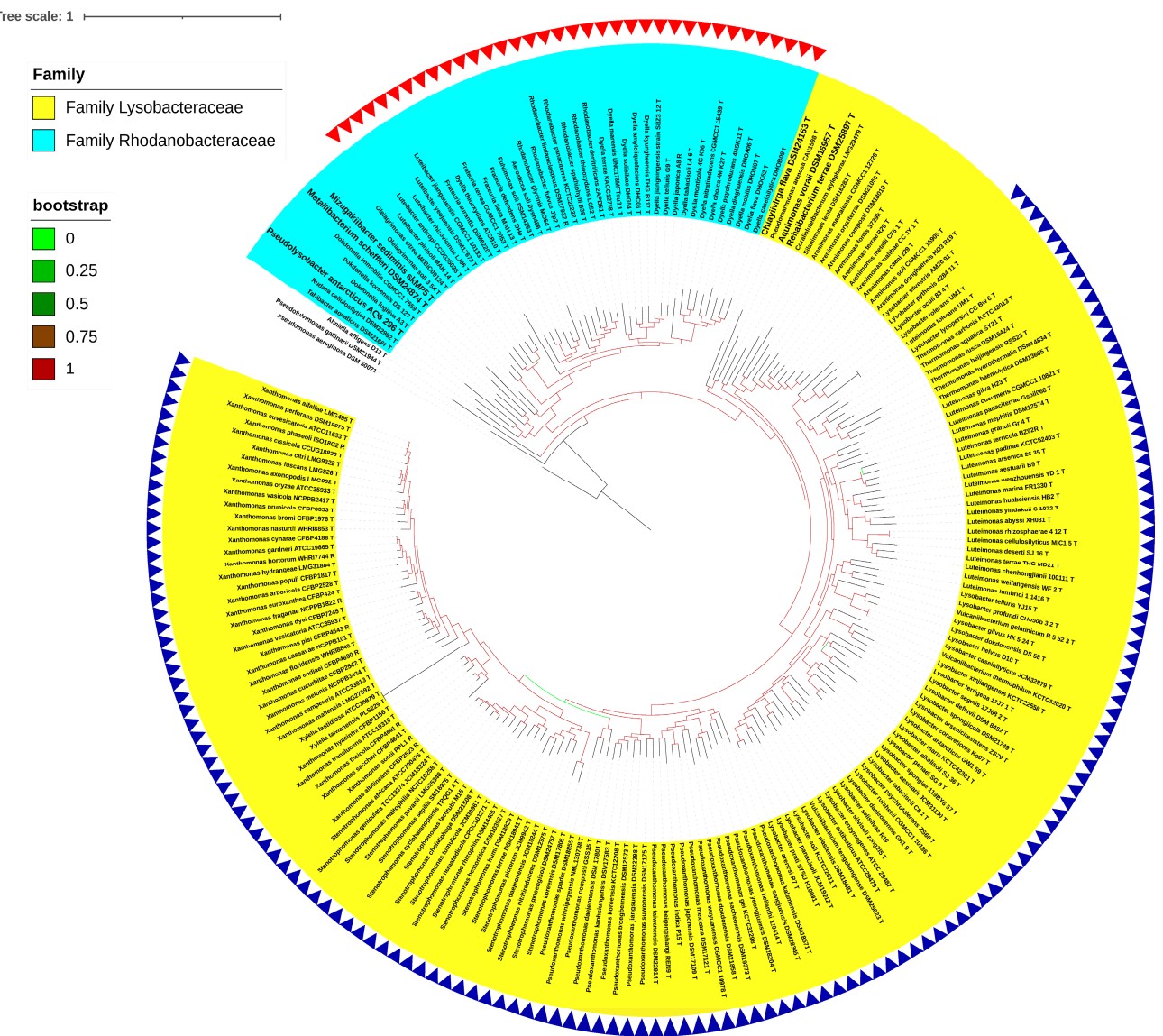

**Figure 3.** PIRATE of type strains. A core-genome-based PIRATE tree is shown with 213 type strains with *Pseudomonas aeruginosa* DSM 50071 00668[T] as an outgroup. The yellow hue represents the *Lysobacteraceae* family, with the blue-colored triangles representing the proposed species belonging to the genus *Xanthomonas*. The *Rhodanobacteracea* family is represented by the sky-blue hue, while the proposed species belonging to the genus *Rhodanobacter* are represented by the red triangles. The bootstrap values are indicated as numbers with color branches.

### 3.3. Phylo-Taxonogenomic Assessment of the Genera of Xanthomonas and Rhodanobacter within Their Respective Families

All three phylogenomic-based trees revealed one major sub-lineage or sub-clade in each of the major clades corresponding to families, i.e., *Lysobactera*ceae (*Xanthomonadaceae*) and *Rhodanobacteraceae* (*Frateuriaceae*). The sub-lineage within *Lysobactera*ceae (*Xanthomonadaceae*) consisted of the genera *Xanthomonas*, *Xylella* [50], *Pseudoxanthomonas* [51], *Stenotrophomonas* [52], *Lysobacter* [53], *Vulcaniibacterium* [12], *Luteimonas* [51], and *Thermomonas*, while the sub-lineage corresponding to *Rhodanobacteraceae* (*Frateuriaceae*) consisted of *Aerosticca* [54], *Fulvimonas* [17], *Rhodanobacter* [16], *Dyella* [18] *Luteibacter* [55], and *Frateuria* [56].

Previously, using deep phylo-taxonogenomics, we reported that *Xanthomonas*, *Xylella*, *Stenotrophomonas*, and *Pseudoxanthomonas* belong to one genus [30]. These four genera are members of a sub-lineage identified in the family *Lysobactera*ceae (*Xanthomonadaceae*). It is

possible that these four genera, along with the other four genera, are all misclassified and in fact belong to one genus. Similarly, six-member genera, i.e., *Aerosticca*, *Fulvimonas*, *Rhodanobacter*, *Dyella*, *Luteibacter*, and *Frateuria*, of the sub-lineage identified in *Rhodanobacteraceae* (*Frateuriaceae*) may also be misclassified and belong to one genus. Hence, it was necessary to confirm the phylogenomic grouping using genome-based taxonogenomic indices.

Accordingly, we used an overall genomic relatedness index (OGRI) for all the type/representative species representing different genera and families of the order. Here, according to the average amino acid identity (AAI) and core average amino acid identity (cAAI) cut-off values of 65% used for delineating a novel genus, the genus boundary for *Xanthomonas* is extended to include eight genera in total (Figures 4 and 5). As indicated earlier, four of these genera were already reported to be synonyms in an earlier in-depth phylo-taxonogenomic study [30]. Here, according to the genus cut-off, all eight of these genera are synonyms of the genus *Xanthomonas*. The 65% AAI threshold also correlates with the whole-genome-based phylogenies, as these genera, with their constituent member species, form a distinct phylogroup within the family (constituent members are indicated by the bold font in the phylogenomic tree) (Figures 1–3). When selecting a cut-off, it is important that the resulting genera be monophyletic [33,34]. Since the evolutionary history of these organisms is unobservable, we relied on sequence similarity and other computational metrics as proxies for evolutionary relationships. Accordingly, the various phylogenies that we present (Figures 1–3) contain clade structures that are consistent with our proposed newly reorganized genera [36,57]. Similarly, according to the genus level cut-off for AAI and cAAI, the genus *Rhodanobacter* includes five more genera, *Aerosticca*, *Fulvimonas*, *Dyella*, *Luteibacter*, and *Frateuria*, that correspond to the sub-lineage identified within the family *Rhodanobacteracea* (Figures 4 and 5). In this case, too, the 65% AAI threshold also correlates with the whole-genome-based phylogenies, as these genera, with their constituent member species, form a distinct phylogroup within the family (constituent members are indicated by the bold font in the phylogenomic tree) (Figures 1–3). Even though it is reported that the vast majority of bacterial intra-genus AAI values are higher than 68%, for a particular or proposed genus under study, the threshold should pass the test of monophyly. Hence, by first establishing the monophyletic nature of the clade or group through deep genome-based phylogenetic trees using two different software (Figures 1–3) and inspecting the AAI of the values (Figures 4 and 5), in the case of *Xanthomonas* and *Frateuria*, we found that the cAAI and AAI cut-off 68% is also applicable to a large number of member species. Hence, considering monophyly test, we propose a cAAI and AAI cut-off of 65% as a threshold in the case of the genera *Xanthomonas* and *Frateuria*.

*3.4. Taxonomic Revision of the Families Lysobacteraceae and Rhodanobacteraceae within the Order Lysobacterales in Light of the Phylo-Taxonogenomic Findings*

Since the genus *Lysobacter* was found to be a synonym of *Xanthomonas*, the genus *Xanthomonas* [58] has precedence over other genera, including *Lysobacter* [53]. Hence, there is a need to explore the validity of the family name, i.e., *Lysobacteraceae*, and the synonym *Xanthomonadaceae*, which is considered illegitimate [1]. Similarly, there is also a need to explore the validity of the order *Lysobacterales* [53] and its synonym, *Xanthomonadales*, which is considered illegitimate [2]. If the International Code of Nomenclature of Prokaryotes (2008 Revision) permits it, then there is a need to reverse the status of, or provide legitimate status to, *Xanthomonadaceae* and *Xanthomonadales*, as proposed in this study. Similarly, there is a need to check the validity of the family names, i.e., *Rhodanobacteraceae* and the genus *Rhodanobacter*, as the genus *Frateuria* was found to be a synonym of *Rhodanobacter* [16], and this genus was proposed earlier than *Rhodanobacter* or any other genera of that family [16].

*3.5. Validating the Deep-Phylo-Taxonogenomic-Based Taxonomic Revelations at the Level of Genus with a Complete 16S rRNA Gene Sequence-Based Threshold*

With the availability of a complete 16S rRNA gene sequence for all the type strains, cut-offs have been proposed for delineating higher taxa. These are 94.5% for a genus, 86.5% for a family, 82.0% for an order, 78.5% for a class, and 75.0% for a phylum [57]. Before

this study, a 16S rRNA sequence identity threshold of 95%, along with a 16S rRNA-based tree, had been used for genus-level delineation [59]. Hence, in the present report, we only propose an emended or expanded description of the genera *Xanthomonas* and *Rhodanobacter* and do not propose any family or emended description of the higher taxa. Hence, we only checked the status of the genera identified in our study. The phylo-taxonogenomic findings (both the AAI threshold of 65% employed and the sub-lineages corresponding to the genera *Xanthomonas* and *Frateuria*), with the 16S rRNA boundaries proposed by Yarza and co-workers for genus delineation, support our revision of the genera *Xanthomonas* and *Rhodanobacter* (Supplementary Table S4). We do not include *Oleiagrimonas soli* in the emended genus *Frateuria* as it does not meet the strict requirement of the monophyly rule [50] and forms an outgroup in the phylogenomic tree(s) (Figures 1–3). However, *Xylella fastidiosa* was included in the emended genus as it meets the strict requirement of the monophyly rule in the phylogenomic tree(s) (Figures 1–3) meeting the 16S rRNA similarity threshold (Figure 6). In fact, *Xylella fastidiosa* clusters with *Xanthomonas* deep within the clade in the phylogenetic trees.

Furthermore, with our proposal for the reclassification of *Lysobacter* as a synonym of *Xanthomonas*, we propose the re-usage of the family name *Xanthomonadaceae* and order name *Xanthomonadales* that are now considered illegitimate (Figure 6).

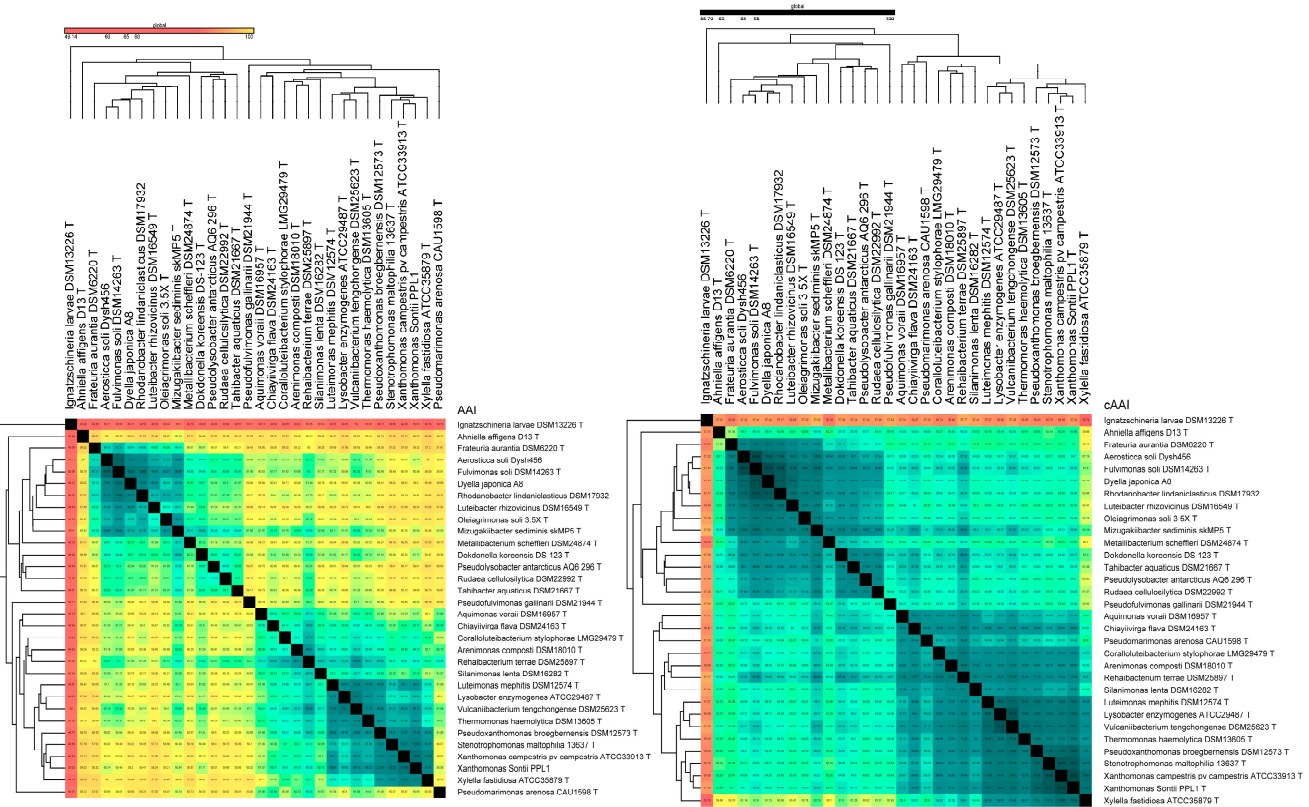

**Figure 4.** AAI and cAAI of type species. Heatmap showing the average amino acid identity (AAI) and core average amino acid identity (cAAI) of 33 type species. The sky-blue color boxes symbolize the *Lysobacteraceae* and the *Rhodanobacteraceae* family.

### 3.6. Emended Description of the Order Xanthomonadales (Saddler and Bradbury 2005a,b) [2]

Synonym: Lysobacterales Christensen and Cook (1978) (Approved Lists 1980)

The order consists of two families, *Xanthomonadaceae* and *Frateuriaceae*. The characteristics of the organisms in the order are as described by Naushad et al. 2014 [1] and Saddler and Bradbury (2005a,b) [60] and based on the phylo-taxonogenomic analysis of the present study. The type genus is *Xanthomonas* (*Xanthomonas campestris* (Pammel 1895) Dowson 1939

(Approved Lists 1980)), as *Lysobacter* (*Lysobacter* Christensen and Cook 1978 (Approved Lists 1980)) was re-classified as a synonym of *Xanthomonas* in the present study.

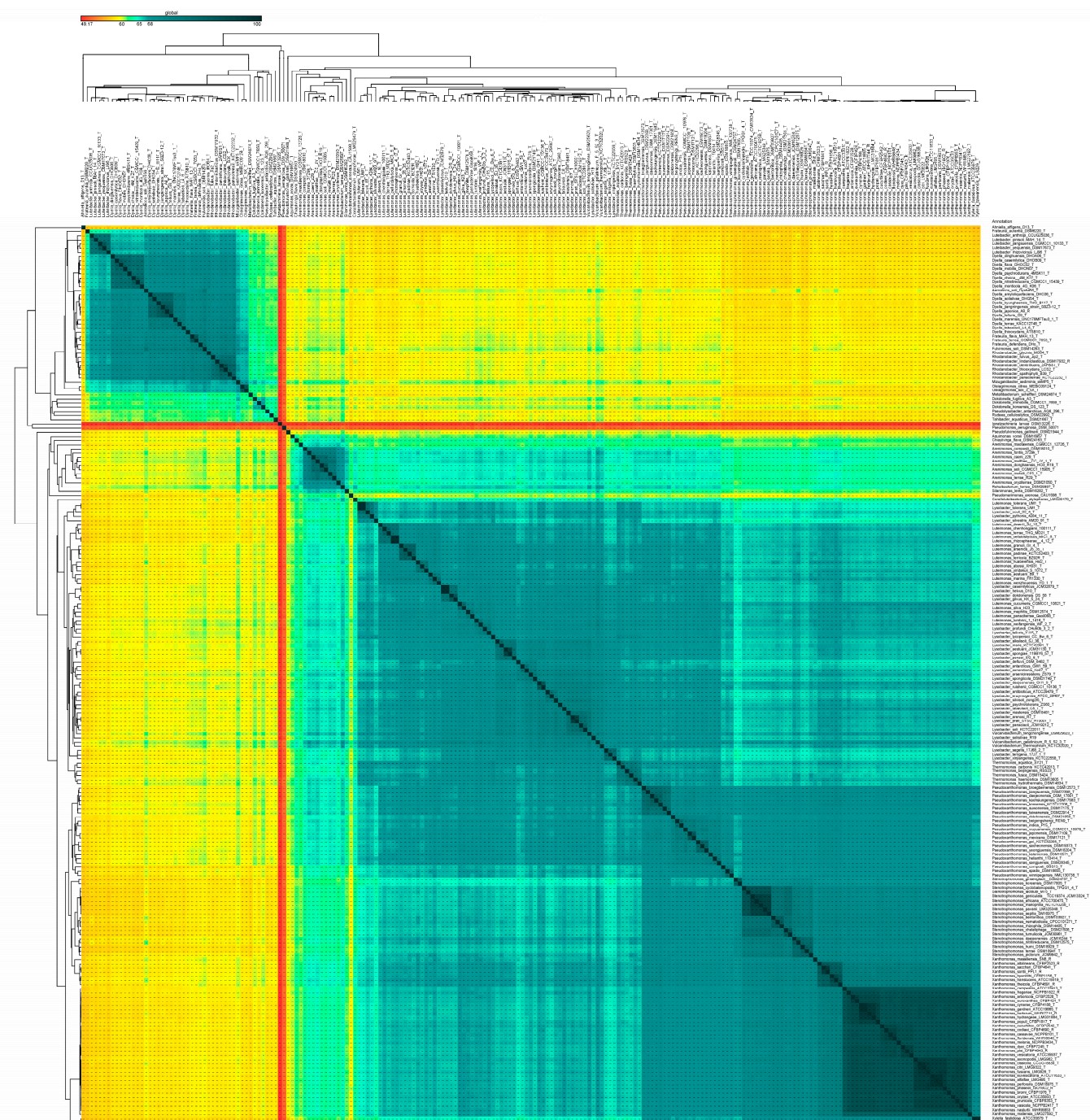

**Figure 5.** AAI of type strains. Heatmap showing average amino acid identity (AAI) amongst 213 type strains. The *Lysobacteraceae* and the *Rhodanobacteraceae* families are represented by the sky-blue boxes.

### 3.7. Emended Description of the Family Xanthomonadaceae Saddler et al. 2005

N.L. fem. n. *Xanthomonas*, type genus of the family; L. fem. pl. n. suff. -aceae, ending to denote a family; N.L. fem. pl. n. *Xanthomonadaceae*, the *Xanthomonas* family.

The description of the family *Xanthomonadaceae* is as given by Saddler et al. 2005 and Christensen and Cook 1978 (Approved Lists 1980), with the following amendments:

Synonym: *Lysobacteraceae* Christensen and Cook 1978 (Approved Lists 1980)

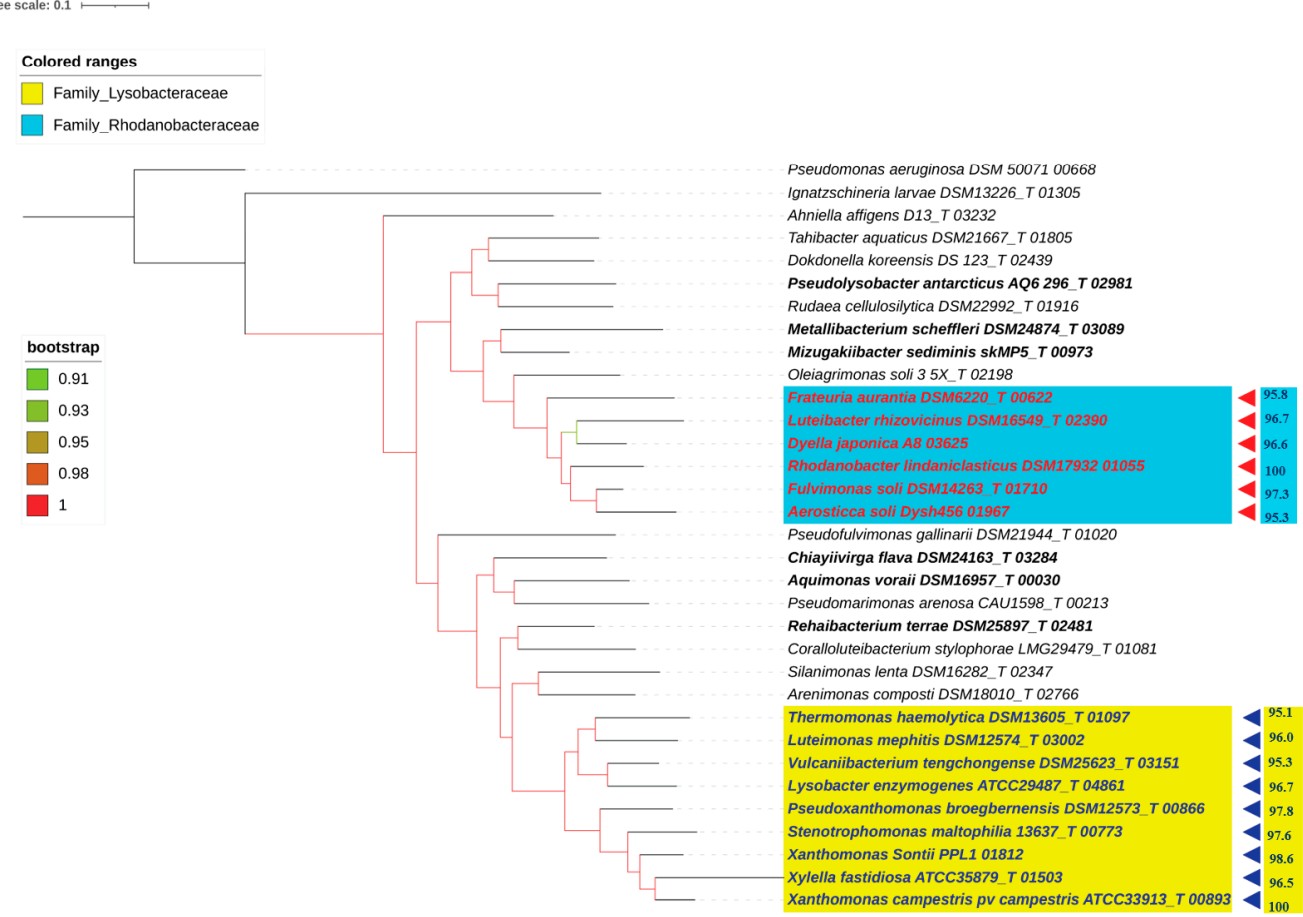

**Figure 6.** PhyloPhlAn tree comprising 33 type species and two outgroups, *Pseudomonas aeruginosa* DSM 50071 00668^T and *Ignatzschineria larvae* DSM13226^T 01305. The 16S rRNA similarity values are shown in front of each genus. Yellow represents the *Lysobacteraceae* family, with blue triangles representing the proposed species in the genus *Xanthomonas*. The sky-blue tint represents the *Rhodanobacteraceae* family, while the red triangles represent the proposed species belonging to the *Rhodanobacter* genus. The bootstrap values are displayed with color branches. Reshuffled genera are shown in bold.

The family is within the order *Xanthomonadales* and includes the genera *Xanthomonas* Dowson 1939 [58], *Coralloluteibacterium* Chen et al. 2018 [61], *Arenimonas* Kwon et al. 2007 [62], *Silanimonas* Lee et al. 2005 [63], and *Rudaea* Weon et al. 2009 [64], with inclusion in *Aquimonas* Saha et al. 2005 [44], *Chiayiivirga* Hsu et al. 2013 [45], *Rehaibacterium* Yu et al. 2013 [46] and exclusion from *Mizugakiibacter* Kojima et al. 2014 [47], *Metallibacterium* Ziegler et al. 2013 [48], and *Pseudolysobacter* Wei et al. 2020 [49]. The member genera of the family were established through the latest deep phylo-taxonogenomic analysis conducted in the present study.

### 3.8. Description of the Family Frateuriaceae fam. nov.

*Frateuriaceae* (Frat.eur'i.a.a,ce'ae N.L. fem. n. *Frateuria* is the type genus of the family; -aceae represents the family; N.L. fem. pl. n. *Frateuriaceae*, the family whose nomenclature type is the genus *Frateuria*).

Synonym: Rhodanobacteraceae Naushad et al. 2015 [1].

The family is within the order *Xanthomonadales*. The proposed family *Frateuriaceae* includes the genera *Frateuria* Swings et al. 1980 [56], *Dokdonella* Yoon et al. 2006 [65], *Pseudolysobacter* Wei et al. 2020 [49], *Rudaea* Weon et al. 2009 [64], and *Tahibacter* Makk

et al. 2014 [66]. *Aquimonas* Saha et al. 2005 [44], *Chiayiivirga* Hsu et al. 2013 [45], and *Rehaibacterium* Yu et al. 2013 [46] are excluded from the previously described family *Rhodanobacteraceae*, while *Mizugakiibacter* Kojima et al. 2014 [47], *Metallibacterium* Ziegler et al. 2013 [48] and *Pseudolysobacter* Wei et al. 2020 are included [49]. The member genera of this family were identified through the latest deep phylo-taxonogenomic analysis conducted in the present study.

*3.9. Emended Description of the Genus Xanthomonas Dowson 1939 (Approved Lists 1980)*

Xan.tho.mo.nas. Gr. masc. adj. xanthos, yellow; L. fem. n. monas, unit, monad; N.L. fem. n. Xanthomonas, yellow monad.

Synonyms: *Xylella, Pseudoxanthomonas, Stenotrophomonas, Lysobacter, Vulcaniibacterium, Luteimonas and Thermomonas.*

The type species is *Xanthomonas campestris* (Pammel 1895) Dowson 1939 (Approved Lists 1980)).

The description is as provided in Naushad et al. and Saddler et al. [1,2] and based on the deep phylo-taxonogenomic analysis conducted in the present study. The genus now includes the previously described genera and their member species, i.e., *Xylella* [50], *Pseudoxanthomonas* [51], *Stenotrophomonas* [52], *Lysobacter* [53], *Vulcaniibacterium* [12], *Luteimonas* [51], and *Thermomonas* [67] (Supplementary Table S1). *Xanthomonas* is a synonym of the previously described genera *Xylella, Pseudoxanthomonas, Stenotrophomonas, Lysobacter, Vulcaniibacterium, Luteimonas*, and *Thermomonas* (Supplementary Table S1).

Emended descriptions of the key member species of the genus *Xanthomonas* are provided below and in Supplementary Table S1.

**Emended description of *Xanthomonas maltophilia*** = *Stenotrophomonas maltophilia* ((Hugh 1981) Swings et al. 1983).
Description as provided in (Hugh 1981) Swings et al. 1983 [56] and the genomic analyses conducted in the present study and a previous study by Bansal et al. 2021 [30].
**Emended description of *Xanthomonas fastidiosa*** = *Xylella fastidiosa* Wells et al. 1987.
Description as provided in Wells et al. 1987 [50] and the genomic analyses conducted in the present study and a previous study by Bansal et al., 2021 [30].
**Emended description of *Xanthomonas broegbernensis*** = *Pseudoxanthomonas broegbernensis* Finkmann et al. 2000.
Description as provided for *Pseudoxanthomonas broegbernensis* in Finkmann et al. 2000 [51] and based on the genomic analyses in the present study and a previous study by Bansal et al., 2021 [30].
**Emended description of the genus *Frateuria* Swing et al. 1980.**
Frat.eur'i.a. N.L. fem. n. Frateuria, named after Joseph Frateur (1903–1974), the eminent Belgian microbiologist. The type species is *Frateuria aurantia* (ex Kondô and Ameyama 1958) Swings et al. 1980 [56].
Synonyms: *Aerosticca, Fulvimonas, Rhodanobacter, Dyella, and Luteibacter.*

The type species is *Frateuria aurantia* (ex Kondô and Ameyama 1958) Swings et al. 1980.

The description is as provided in Naushad et al. 2015 [1] and based on the deep phylo-taxonogenomic analysis conducted in the present study. The genus now includes the previously described genera and their member species, i.e., the genus now includes genus *Aerosticca, Fulvimonas, Rhodanobacter, Dyella, Luteibacter*, and *Frateuria* (Supplementary Table S2).

The emended descriptions of the member species of the genus *Frateuria* are provided in Supplementary Table S2.

**Supplementary Materials:** The following supporting information can be downloaded at: https://www.mdpi.com/article/10.3390/taxonomy3040026/s1, Table S1: List of proposed species of the genus Xanthomonas; Table S2: List of proposed species of the genus *Rhodanobacter*; Table S3: Metadata showing the list of all of the genomes (213) and their accessions number used in this study belonging to the type species and type strains of the order *Lysobacterales* (*Xanthomonadales*); Table S4: Metadata showing 16S rRNA % similarity values using the Phylogeny server (DSMZ).

**Author Contributions:** K.B., S.K. and A.S.: data analysis and drafting of the manuscript. A.C.: metadata preparation. P.B.P. conceived and participated in the planning, design, and drafting of the manuscript. All authors have read and agreed to the published version of the manuscript.

**Funding:** This work was supported by an institutional project grant provided to MTCC and CSIR MPL065 provided to PBP from CSIR.

**Data Availability Statement:** Not applicable.

**Conflicts of Interest:** The authors declare no conflict of interest.

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
