# Peer review of "Redefining the Taxonomic Boundaries of Genus Xanthomonas"

_2673-6500, doi:10.3390/taxonomy3040026_

Round 1

Reviewer 1 Report

This work comprises deep phylotaxonogenomics of the members of the order Lysobacterales, focusing the work on the current taxonomic boundaries of the genera belonging to Lysobacteraceae and Rhodanobacteraceae. Whole genome-based phylogeny including the type species of the genera reported in Lysobacterales revlealed two clades, then whole genome-based phylogeny using 213 type stains also revealed two major clades, which was confirmed using conserved genes. Hence, propose emended description of genera Xanthomonas and Rhodanobacter.

Comments

Figures

- It is hard to distinguish the size of the triangle in the Figures , please, put colors or different shapes to facilitate reading.

- Please correct Figure: Scientific names in italics. and "T" in superscript when referring to type species.

Introduction

1. 16S in some cases is not in uppercase. Please check throughout the text.

2. Check that references are correctly merged, for example: [8] [9, 10] should be [8-10].

3. Pseudomarimonas, Lysobacter, and Vulcaniibacterium etc. consisting -> delete "etc."

4. L. enzymogenes C3 -> Remove italics in C3.

5. The species like 3.5XT of -> Indicate the full name of the species.

6. Authors refer to several methods for genus and species delimitation, but authors do not mention digital DNA-DNA hybridization in the introduction. Please refer to this method for delineating species.

7. delineation of high taxa especially genus -> taxa, especially

8. was reported an variant Xanthomonas -> an?

9. content [26] [8, 28]. -> merge reference numbers.

10. Authors do not mention how core genes were defined in those cited methods [26, 27, 31].

11. phylotaxonomic- genomic -> phylotaxonomic-genomic.

Methods

12. In methods, section Genome access and quality assessment. Please indicate how many species passed this filter and indicate whether some species were not included in the analysis. Provide the stats in a Supp. Table.

13. what does the sentence "core genome with a high level of robustness." mean? Could you please elaborate more on this.

Result and Discussion

14. Authors mention that analysed 213 genomes Out of how many type species described?

15. First parraghraph in section "Phylogenomic evaluation of the families and their boundaries within the order Lysobacterales (Xanthomonadales)" can be reduced as a Table.

16. A Whole-genome based phylogeny using PhyloPhlAn is not specified in the methods. In the methods that authors specify analyses of core genomes.

17. The separation of Xanthomonas species seems arbitrary rather than a result of the analysis conducted. Please, authors be more specific.

18. After unifying the genera of Xanthomonas or Rhodanobacter with other related genera, did the authors perform a dDDH to identify possible subspecies? If not, please perform dDDH.

19. In Figure 2, remove italics in "family".

20. Please describe the methodology for genomic relatedness index (OGRI).

21. Figure 6. Please check the title and label the columns.

22. 16S rRNA gene cutoff should be complemented with dDDH in order to identify possible subspecies of the genera analysed.

Author Response

Manuscript number: Taxonomy-2276947

Title: Redefining the taxonomic boundaries of genus Xanthomonas

Authors: Kanika Bansal1#, Sanjeet Kumar1#, Anu Singh1#, Arushi Chaudhary1 and Prabhu B. Patil1*

We thank the reviewers for their valuable comments, suggestions and feedback on our manuscript towards the improvement of the manuscript. We have addressed the comments and incorporated the suggestions of reviewers as detailed below. Reviewers’ comments are in blue and our responses are in black.

Comments and Suggestions for Authors

Reviewer 1

This work comprises deep phylotaxonogenomics of the members of the order Lysobacterales, focusing the work on the current taxonomic boundaries of the genera belonging to Lysobacteraceae and Rhodanobacteraceae. Whole genome-based phylogeny including the type species of the genera reported in Lysobacterales revealed two clades, then whole genome-based phylogeny using 213 type stains also revealed two major clades, which was confirmed using conserved genes. Hence, propose emended description of genera Xanthomonas and Rhodanobacter.

Comments

Figures

- It is hard to distinguish the size of the triangle in the Figures, please, put colors or different shapes to facilitate reading.

We thank you for the comment. We have now provided bootstrap values with color branches instead of triangles in revised manuscript

- Please correct Figure: Scientific names in italics. and "T" in superscript when referring to type species.

We thank you for the comment. We have made changes figure in revised manuscript. Since in the iTOL software, we were not able to make T in superscript we have indicated as _T

Introduction

  1. 16S in some cases is not in uppercase. Please check throughout the text.

We thank you for the comment. We have made all the changes in revised manuscript

  1. Check that references are correctly merged, for example: [8] [9, 10] should be [8-10].

We thank you for the comment. We have made changes in revised manuscript (line 46)

  1. Pseudomarimonas, Lysobacter, and Vulcaniibacterium etc. consisting -> delete "etc."

We thank you for the comment. We deleted in revised manuscript (line 49)

  1. enzymogenes C3 -> Remove italics in C3.

We thank you for the comment. We corrected in revised manuscript (line 55)

  1. The species like 3.5XT of -> Indicate the full name of the species.

We thank you for the comment. We have given full name in revised manuscript (line 60)

  1. Authors refer to several methods for genus and species delimitation, but authors do not mention digital DNA-DNA hybridization in the introduction. Please refer to this method for delineating species.

We thank the reviewer for the comment and suggestion. As our primary aim was to investigate genus level classifications and above. Since dDDH is not used investigate genera and above taxa but for species/sub-species delineations, we have not used in our study.

  1. delineation of high taxa especially genus -> taxa, especially

We have made the correction in in the revised manuscript

  1. was reported an variant Xanthomonas -> an?

We have made the correction in in the revised manuscript

  1. content [26] [8, 28]. -> merge reference numbers.

We thank you for the comment. We have merged in revised version

  1. Authors do not mention how core genes were defined in those cited methods [26, 27, 31].

We thank the reviewer for the comment. We have now mentioned the definition in the revised manuscript and cited the correct references as 28 and 33 references. (line: 86-87)

  1. phylotaxonomic- genomic -> phylotaxonomic-genomic.

We have made the correction in in the revised manuscript

Methods

  1. In methods, section Genome access and quality assessment. Please indicate how many species passed this filter and indicate whether some species were not included in the analysis. Provide the stats in a Supp. Table.

We thank the reviewer for the comment and suggestion. The genomes of the species that did not pass the checkM filter is provided in the figshare link, https://figshare.com/s/d44fbd32de1514614610

  1. what does the sentence "core genome with a high level of robustness." mean? Could you please elaborate more on this.

We thank the reviewer for the comments. We have now deleted the sentence as it is not making any sense.

Result and Discussion

  1. Authors mention that analysed 213 genomes Out of how many type species described?

The genomes of the species that did not pass the filter is provided in the figshare link, https://figshare.com/s/d44fbd32de1514614610

  1. First paragraph in section "Phylogenomic evaluation of the families and their boundaries within the order Lysobacterales (Xanthomonadales)" can be reduced as a Table.

We thank the reviewer for the suggestion. We have provided table in this regard but also keeping the first paragraph as this will helping us to meet the number of words required for the article in this journal and as mentioned in the decision letter.

  1. A Whole-genome based phylogeny using PhyloPhlAn is not specified in the methods. In the methods that authors specify analyses of core genomes.

We thank the reviewer for the comment. We find that we have mentioned the PhyloPhlAn in the methods (line 121-124)

  1. The separation of Xanthomonas species seems arbitrary rather than a result of the analysis conducted. Please, authors be more specific.

We thank the reviewer for the comment. Since we are focussing on classifications at the level of genus and above, we have not investigated the species classifications and used the existing classifications which are mostly accurate.

  1. After unifying the genera of Xanthomonas or Rhodanobacter with other related genera, did the authors perform a dDDH to identify possible subspecies? If not, please perform dDDH.

We thank the reviewer for the comment and suggestion. As our primary aim was to investigate above species level or genus level classifications and above. Since dDDH is not used investigate genera and above taxa but for species/sub-species delineations, we have not used in our study.

  1. In Figure 2, remove italics in "family".

We have made the correction in the revised manuscript.

  1. Please describe the methodology for genomic relatedness index (OGRI).

We thank you for the comment. In the modified manuscript we have described the methodology for OGRI (line 131-136 and line 208-212)

  1. Figure 6. Please check the title and label the columns.

We thank you for the comment. We have made the correction in the revised manuscript.

  1. 16S rRNA gene cut-off should be complemented with dDDH in order to identify possible subspecies of the genera analysed.

We thank the reviewer for the comment and suggestion. As our primary aim was to investigate above species level or genus level classifications and above. Since dDDH is not used investigate genera and above taxa but for species/sub-species delineations, we have not used in our study.

Reviewer 2 Report

Summary of Article

Previously [8], the authors have presented compelling computational evidence that the genera Xanthomonas, Xylella, Pseudoxanthomas, and Stenotrophomonas represent a single genus, Xanthomonas. In this paper, the authors apply their computational approach to the entire order, Lysobacterales. From their results, they argue for a number of major changes to the order, including,

  • Xanthomonas, Xylella, Pseudoxanthomonas, Stenotrophomonas, Lysobacter, Vulcaniibacterium, Luteimonas, and Thermomonas into a single genus.

  • With Xanthomonas and Lysobacter merged a single genus, the name Xanthomonas has precedence, so Lysobacteraceae would become Xanthomonadaceae, and Lysobacterales would become Xanthomonadales.

  • “Similarly, six-member genera i.e., Aerosticca, Fulvimonas, Rhodanobacter, Dyella, Luteibacter, and Frateuria […] may also are misclassified and belong to one genus” to be named Frateuria.

  • As a result, over 140 species would be renamed (comb. nov.), e.g., Xylella fastidiosa becomes Xanthomonas fastidiosa.

The authors go further and observer that “the monophyly test as revealed by major sub-lineages corresponding to these genera indicate that we have been conservative in selecting the AAI and 16s rRNA cut-offs.” They go on to observe that the methods and cutoffs that they have used would suggest the merging of the families Lysobacteraceae and Rhodanobacteraceae as well. However, they propose waiting for the “[d]iscovery of additional families and their taxonomic/phylogenetic depth” before making that determination.

High-level comments

I find this paper and their previous paper [8] absolutely mind-blowing. The authors show that purely computational methods can be used to think about taxonomic “rationalization” at the genus and family level, and beyond. However, I believe that this manuscript suffers from two major issues:

  1. The authors select 65% similarity cutoff for genus delineation. While they point out that other authors use cutoffs in the range of 60%-80%, these authors fail to consider or discuss using a high cutoff in order to produce monophyletic genera that require fewer changes to existing taxonomy.

  2. Their use of 16s similarity is hopeless. Their method and results would argue for combining E.coli, P.aeruginosa, and X.campestris into the same family (at worst) or order (at best), a fact that they observe themselves. The authors fail to realize that breadth of the changes suggested by their results undermines their use to support the changes that they authors do discuss. Plus, there is something funny about the method used to measure similarity. However, fixing the method does not fully address this issue.

Each of these issues is discussed in more detail in “Major Issue 1” and “Major Issue 2”.

I would very much like to see the authors address these two major issues and resubmit. I believe that their work is very exciting and is likely to provoke much discussion. Perhaps the authors could, instead of vaguely implying that more changes than those mentioned should be considered eventually, rework the paper to discuss the limits of similarity-based methods and arbitrary cutoffs for evaluating the rationality of existing higher taxa. The 16s material could easily be reworked in this manner.

Specific comments

Abstract

“with prominent being Lysobacter” - awkward. Perhaps, “with a notable being Lysobacter”.

“of related genus” - should be “of the related genus”

“taxonomy based comparative studies” - should be “taxonomy-based comparative studies”

1. Introduction

“The species like 3.5X(T) of genus Oleiagrimonas play a special role” - two problems: missing commas and “3.5X(T)” is not a species name. Perhaps, “Species, like Oleiagrimonas soli, play a special role”.

“The first and major whole-genome phylotaxonomic- genomic based study of the order”. Lots of missing commas; extra space after “phylotaxonomic-”.

2. Materials and Methods

Genome access and quality assessment

Please mention that Table S3 contains (I think!) the list of all of the genomes and their accesions used in this study.

“Genomes of the strains used in the study was obtained from NCBI […] and GTDB […]” - Why both repositories? Did it matter? Should you document which come from which?

“The genomes that passed the QC (< 5% contamination and < 95% completeness) were considered […]” - You need to include a list of the genomes, i.e., type strains, i.e., species that were not included in your study. Does Table S3 represent only the genomes that were considered? Please clarify.

Phylogenomics investigation …

“using PIRATE [34].” - I don’t think that [34] is the correct reference. I believe that it should be doi:10.1093/gigascience/giz119

Also, which version of PIRATE did you use?

“The 16s rRNA phylogeny and similarity was generated using DSMZ Phylogeny server (https://ggdc.dsmz.de/phylogeny-service.php).” - There is something funny about the results produced by this server. See the comments below I have labeled “Major Issue 2”.

3. Results and discussion

Phylogenomic evaluation …

“A total of 213 genomes […] are summarised in the table (supplementary Table 3).” - Please clarify: how many type strains did not have available genomes? Also, how many genomes were not used because they failed the CheckM test?

“confirmed using a core genome-based tree”. It would be more clear if it was stated as “confirmed using a core pangenome-based tree”.

“pan genome”. Should be “pangenome”.

“Pseudofulvimonas gallinarii and Ahniella affigens formed outgroup for these families respectively.” Are you suggesting they fall in families other than Lysobacteraceae or Rhodanobacteraceae? Please clarify.

Phylo-taxonogenomic assessment …

“Genome based trees” should be “Genome-based trees”.

“in the each of the major clade” should be “in each of the major clades” or “in each major clade”.

“cut-off values of 65% used for delineating novel genus”. Please the comments labelled “Major Issue 1”.

“All global and targeted studies on genus delineation reiterate that monophyly is more essential requirement than the mere fulfilment of a strict threshold [33, 34]. Accordingly results in this study also correlate with the whole genome-based phylogeny, as these genera with their constituent member species form a distinct phylogroup within the family (constituent members indicated by the bold font in the phylogenomic tree) (Figure 1, 2 and 3).” I find this very confusing. I think what you are trying to say is, “When selecting a cutoff, it is important that resulting genera by monophyletic. Since the evolutionary history of these organisms is unobservable, we rely on sequence similarity and other computational metrics as proxies for evolutionary relationships. Accordingly, the various phylogenies that we have presented (Figures 1-5) contain clade structures that are consistent with our proposed the newly reorganized genera.”

“(n=33)”. This is a very minor point: in the paper you use the phrase, “type strain” in two senses, species-level “type strain” and genus-level “type strain”. Your usage is correct, but since you are working with both sets of strains (n=213 and n=33), you might want to use terms that make it clear which set you are talking about each time.

Taxonomic validity of …

Are you discussing the validity of the families or the names of the families?

“If the code permit” should be “If the Code permits”. Also, I belive that this is the first reference to “the Code”, so it should be, “If the International Code of Nomenclature of Prokaryotes (2008 Revision) [doi:10.1099/ijsem.0.000778] permits”, or similar.

Validating the deep-phylotaxonogenomic …

“While the advent of genomics era has allowed to propose taxonomic cut-offs for species and genera, robust genomic indices for higher taxa like families and above are not yet available.” Awkward. Perhaps instead, “The advent of genomics has led to quasi-standard genomics-based cut-offs for species (95% ANI) and genera (60%-80% AAI), there are no widely accepted cutoffs for higher taxa.”

“These are: 94.5% for genus, 86.5% for family, 82.0% for order, 78.5% for class and 75.0% for phylum [34].” See the comments I have labeled “Major Issue 2”.

Oleiagrimonas soli and Xyllela fastidosa are boundary cases in your analysis. Discuss.

“In fact, the monophyly test as revealed by major sub-lineages corresponding to these genera indicate that we have been conservative in selecting the AAI and 16s rRNA cut-offs.” The is unnecessarily vague. Examining Supp table 4, I can imagine what you mean, but it either be explicit or delete the point. Also, see “Major Issue”.

“saturation through discoveries”. I have no idea what this means. Please restate.

Figures

Figure 1.

The image is fuzzy, please provide a higher resolution image

Triangles of various size are used to represent the bootstrap values on the tree. However, the sizes of the triangles are indistinguishable. Please provide the numeric values or use colored triangles to enable the reader to discern these values.

“blue triangles representing species in the genus Xanthomonas.”, “the red triangles represent the Rhodanobacter genus.” The caption text should be changed to indicate that the authors are proposing that these species be places in the genera Xanthomonas and Rhodanobacter.

Figure 2.

Same as Figure 1: The image is blurry. The labels are unreadable. The triangles are indistinguishable. In fact, the triangles are so small they are invisible. Move a high-res image to Supplemental?

The caption: Ditto figure 1 - the placement of species in genera is proposed.

Figure 3.

Ditto Figures 1 and 2.

Figure 4.

Ditto Figures 1-3. Also, the values on the heatmap scale are illegible. Furthermore, is there a good reason why the AAI and cAAI heatmaps order the strains differently?

Figure 5.

Ditto Figures 1-4. I think that it is extremely unlikely that you will be able to make this figure legible while it is in the main text. Please consider including a (very!) high-resolution image in Supplemental.

Figure 6.

Fuzzy. Please provide a sharper image.

Why was 88.07 for P.aeruginosa omitted?

Supplemental Table 3

I note the following genera that you propose to move between families:

Rhodanobacteraceae -> Lysobacteraceae

  • Aquimonas voraii DSM 16957_T
  • Chiayiivirga flava DSM 24163_T
  • Rehaibacterium terrae DSM 25897_T

Lysobacteraceae -> Rhodanobacteraceae

  • Metallibacterium scheffleri DSM 24874_T
  • Mizugakiibacter sediminis skMP5_T
  • Pseudolysobacter antarcticus strain:AQ6-296_T

I do not find discussion of any of these in the paper. Find rectify this.

Major Issue 1

“cut-off values of 65% used for delineating novel genus”.

Why/how was this cutoff chosen? Previously you reported works that use cutoffs ranging from range 60%-80%. The choice of this cutoff is one of the most important details of this paper, and there is no detailed discussion about how you arrive at a cutoff value of 65%. It would appear to be arbitrary, which significantly undermines your stated claim that the species and genera need to be reorganized the status quo.

Consider, is 65% the only cutoff less than 80% that satisfies the monophyletic requirement that you have stated? If it is, then please demonstrate this, because I believe that it would be the most convincing evidence to support your reorganization. If it is not, then shouldn’t a more conservative cutoff be selected that results in fewer changes to existing taxonomy?

Major Issue 2

“These are: 94.5% for genus, 86.5% for family, 82.0% for order, 78.5% for class and 75.0% for phylum [34].”

While I accept these are the thresholds used in [34], I would claim that these thresholds, when applied using your own method, give nonsensical results. Your results (Suppl table 4) and these thresholds would place P.aeruginosa and X.campestris not only in the same order, but in the same family. You more or less state this in the discussion.

In order illustrate the problem, I used the GGDC server to compute similarity measures for the 16s sequences downloaded from LPSN for E.coli, P.aeruginosa, and X.campestris, which are all in same class but each in a different order. Here are the results:

  ecoli paeruginosa xcampestris
ecoli 100 88.45 88.7
paeruginosa 88.45 100 88.07
xcampestris 88.7 88.07 100

Again, your method would place all of these in the same family.

A small improvement would be to use another method for computing 16s similary. Here are two possibilities:

NCBI “blastn”:

  ecoli paeruginosa xcampestris xfastidiosa
ecoli 100 85.135 84.963 84.019
paeruginosa 85.135 100 85.267 84.913
xcampestris 84.963 85.267 100 95.678
xfastidiosa 84.019 84.913 95.681 100

“water” (pairwise Smith-Waterman, found in the EMBOSS package):

  ecoli paeruginosa xcampestris xfastidiosa
ecoli 100 84.4 83.7 82.2
paeruginosa 84.4 100 84.6 83.9
xcampestris 83.7 84.6 100 95.4
xfastidiosa 82.2 83.9 95.4 100

Either of these gives much more conservative results than GGDC, and these results + the [34] cutoffs give results much closer to the status quo. E.g., for “water”, each pair in the same…

  ecoli paeruginosa xcampestris xfastidiosa
ecoli strain order order order
paeruginosa order strain order order
xcampestris order order strain genus
xfastidiosa order order genus strain

This is better, but still nonsense: E.coli, P.aeruginosa, and X.campestris are placed in the same order.

“However, the same was not the case with the families know in order Lysobacteraceae. In the absence of genomic indices for taxa above genus level, the proposed 16s rRNA boundaries question of the existence or earlier proposal of family Lysobacteraceae and Rhodanobacteraceae as the cut-off of the members are way above 86.7%[34].”

Your 16s results and discussion would lead the reader believe that you are suggestung reconsidering the entire taxonomy of class Gammaproteobacteria. I would respectfully suggest that your approach (GGDC + the cutoffs from [34]) should be reconsidered instead. Everything in this manuscript related to 16s similarity needs to be redone - methods, results, and discussion, all of it. Your results would have well-established orders of Gammaproteobacteria devolving into a single family (at worst) or order (at best), which is unreasonable. Therefore, these (unreasonable) results do not support your (more reasonable) adjustments within Lysobacterales (Xanthomonadales).

Author Response

Manuscript number: Taxonomy-2276947

Title:
Redefining the taxonomic boundaries of genus Xanthomonas

Authors: Kanika Bansal1#, Sanjeet Kumar1#, Anu Singh1#, Arushi Chaudhary1 and Prabhu B. Patil1*

We thank the reviewer for their valuable comments, suggestions and feedback on our manuscript. We have addressed the comments and incorporated all the suggestions of reviewers, which will definitely make the manuscript clearer and more improved. Reviewers’ comments are in blue and our responses are in black.

Reviewer 2

Previously [8], the authors have presented compelling computational evidence that the genera Xanthomonas, Xylella, Pseudoxanthomas, and Stenotrophomonas represent a single genus, Xanthomonas. In this paper, the authors apply their computational approach to the entire order, Lysobacterales. From their results, they argue for a number of major changes to the order, including,

  • Xanthomonas, Xylella, Pseudoxanthomonas, Stenotrophomonas, Lysobacter, Vulcaniibacterium, Luteimonas, and Thermomonas into a single genus.
  • With Xanthomonas and Lysobacter merged a single genus, the name Xanthomonas has precedence, so Lysobacteraceae would become Xanthomonadaceae, and Lysobacterales would become Xanthomonadales.
  • “Similarly, six-member genera i.e., Aerosticca, Fulvimonas, Rhodanobacter, Dyella, Luteibacter, and Frateuria […] may also are misclassified and belong to one genus” to be named Frateuria.
  • As a result, over 140 species would be renamed (comb. nov.), e.g., Xylella fastidiosa becomes Xanthomonas fastidiosa.

The authors go further and observer that “the monophyly test as revealed by major sub-lineages corresponding to these genera indicate that we have been conservative in selecting the AAI and 16s rRNA cut-offs.” They go on to observe that the methods and cutoffs that they have used would suggest the merging of the families Lysobacteraceae and Rhodanobacteraceae as well. However, they propose waiting for the “[d]iscovery of additional families and their taxonomic/phylogenetic depth” before making that determination.

High-level comments

I find this paper and their previous paper [8] absolutely mind-blowing. The authors show that purely computational methods can be used to think about taxonomic “rationalization” at the genus and family level, and beyond. However, I believe that this manuscript suffers from two major issues:

  1. The authors select 65% similarity cutoff for genus delineation. While they point out that other authors use cutoffs in the range of 60%-80%, these authors fail to consider or discuss using a high cutoff in order to produce monophyletic genera that require fewer changes to existing taxonomy.
  2. Their use of 16s similarity is hopeless. Their method and results would argue for combining E.coli, P.aeruginosa, and X.campestris into the same family (at worst) or order (at best), a fact that they observe themselves. The authors fail to realize that breadth of the changes suggested by their results undermines their use to support the changes that they authors do discuss. Plus, there is something funny about the method used to measure similarity. However, fixing the method does not fully address this issue.

Each of these issues is discussed in more detail in “Major Issue 1” and “Major Issue 2”.

I would very much like to see the authors address these two major issues and resubmit. I believe that their work is very exciting and is likely to provoke much discussion. Perhaps the authors could, instead of vaguely implying that more changes than those mentioned should be considered eventually, rework the paper to discuss the limits of similarity-based methods and arbitrary cutoffs for evaluating the rationality of existing higher taxa. The 16s material could easily be reworked in this manner.

We thank the reviewer for the critical comments and also valuable suggestions. As indicated in the later section where these points are highlighted in detail, we have made required modifications in the revised manuscript.

Specific comments

Abstract

“with prominent being Lysobacter” - awkward. Perhaps, “with a notable being Lysobacter”.

We thank you for the comment. We have corrected the statement in revised manuscript in abstract (line 20)

“of related genus” - should be “of the related genus”

We thank you for the comment. Correction has been done (line 22)

“taxonomy based comparative studies” - should be “taxonomy-based comparative studies”

We thank you for the comment. Changed to “taxonomy-based comparative studies” (line 23)

  1. Introduction

“The species like 3.5X(T) of genus Oleiagrimonas play a special role” - two problems: missing commas and “3.5X(T)” is not a species name. Perhaps, “Species, like Oleiagrimonas soli, play a special role”.

We thank the reviewer for the comment and suggestion. We have now made the correction in revised manuscript (line 60)

“The first and major whole-genome phylotaxonomic- genomic based study of the order”. Lots of missing commas; extra space after “phylotaxonomic-”.

We thank the reviewer for the comment. We have now modified the sentence in the revised manuscript to “In an earlier study, by carrying out a deep phylo-taxonogenomic investigation of the order Lysobacterales, we reported major reshufflings at the family level, apart from revealing the boundary of the order and its outliers.” (results and discussion)

  1. Materials and Methods

Genome access and quality assessment

Please mention that Table S3 contains (I think!) the list of all of the genomes and their accesions used in this study.

We thank you for the comment. We have mentioned accession/ GCA numbers information in supplementary table 3 in revised manuscript

“Genomes of the strains used in the study was obtained from NCBI […] and GTDB […]” - Why both repositories? Did it matter? Should you document which come from which?

We thank reviewer for the comment. We have used genomes from NCBI and mentioned the same in the revised manuscript

“The genomes that passed the QC (< 5% contamination and < 95% completeness) were considered […]” - You need to include a list of the genomes, i.e., type strains, i.e., species that were not included in your study. Does Table S3 represent only the genomes that were considered? Please clarify.

We thank the reviewer for the comment and suggestion. While we included the genome of all the valid published genus level type strains as they passed the filter. With respect to species level type strains, we included only 213 genomes as they passed the filter. The genomes of the strains that did not pass the filter is provided in the figshare link https://figshare.com/s/d44fbd32de1514614610  as table (supplementary table 5).

Phylogenomics investigation …

“using PIRATE [34].” - I don’t think that [34] is the correct reference. I believe that it should be doi:10.1093/gigascience/giz119

Also, which version of PIRATE did you use?

We have put the correct reference for PIRATE in revised manuscript now (line 127). We have used

Version 1.1.1 by Gabriel Barone

“The 16s rRNA phylogeny and similarity was generated using DSMZ Phylogeny server (https://ggdc.dsmz.de/phylogeny-service.php).” - There is something funny about the results produced by this server. See the comments below I have labeled “Major Issue 2”.

  1. Results and discussion

Phylogenomic evaluation …

“A total of 213 genomes […] are summarised in the table (supplementary Table 3).” - Please clarify: how many type strains did not have available genomes? Also, how many genomes were not used because they failed the CheckM test?

We thank the reviewer for the comment and suggestion. While we included the genome of all the valid published genus level type strains as they passed the filter. With respect to species level type strains, we included only 213 genomes as they passed the filter. The genomes of the strains that did not pass the filter is provided in the figshare link https://figshare.com/s/d44fbd32de1514614610 as table (supplementary table 5).

“confirmed using a core genome-based tree”. It would be more clear if it was stated as “confirmed using a core pangenome-based tree”.

We thank you for the comment. We have changed to- “confirmed using a core pangenome-based tree”.

“pan genome”. Should be “pangenome”.

We thank you for the comment. We have corrected to “pangenome” in revised manuscript

“Pseudofulvimonas gallinarii and Ahniella affigens formed outgroup for these families respectively.” Are you suggesting they fall in families other than Lysobacteraceae or Rhodanobacteraceae? Please clarify.

We thank reviewer for the comment. In the current study we are not proposing them as distinct families and at the same time we are excluding from known families based on wgs trees and 16s rRNA tree. However, we are including in the order.

Phylo-taxonogenomic assessment …

“Genome based trees” should be “Genome-based trees”.

We thank you for the comment. we have corrected to “Genome-based trees” in revised manuscript

“in the each of the major clade” should be “in each of the major clades” or “in each major clade”.

We thank you for the comment. We have corrected to “in each of the major clades” in revised manuscript

“cut-off values of 65% used for delineating novel genus”. Please the comments labelled “Major Issue 1”.

“All global and targeted studies on genus delineation reiterate that monophyly is more essential requirement than the mere fulfilment of a strict threshold [33, 34]. Accordingly results in this study also correlate with the whole genome-based phylogeny, as these genera with their constituent member species form a distinct phylogroup within the family (constituent members indicated by the bold font in the phylogenomic tree) (Figure 1, 2 and 3).” I find this very confusing. I think what you are trying to say is, “When selecting a cutoff, it is important that resulting genera by monophyletic. Since the evolutionary history of these organisms is unobservable, we rely on sequence similarity and other computational metrics as proxies for evolutionary relationships. Accordingly, the various phylogenies that we have presented (Figures 1-5) contain clade structures that are consistent with our proposed the newly reorganized genera.”

We thank reviewer for the comment and suggestion. We have modified the para as mentioned as per the suggestion.

“When selecting a cut off, it is important that resulting genera by monophyletic (33,34). Since the evolutionary history of these organisms is unobservable, we rely on sequence similarity and other computational metrics as proxies for evolutionary relationships. Accordingly, the various phylogenies that we have presented (Figures 1-5) contain clade structures that are consistent with our proposed the newly reorganized genera.”

“(n=33)”. This is a very minor point: in the paper you use the phrase, “type strain” in two senses, species-level “type strain” and genus-level “type strain”. Your usage is correct, but since you are working with both sets of strains (n=213 and n=33), you might want to use terms that make it clear which set you are talking about each time.

We thank for the reviewer for the comment and suggestion. We have tried to clear the same in the revised supplementary table no 3.

Taxonomic validity of …

Are you discussing the validity of the families or the names of the families?

We thank the reviewer for the comment. We have now modified the sub-title to “Taxonomic validity of the names of the families ….”

“If the code permit” should be “If the Code permits”. Also, I belive that this is the first reference to “the Code”, so it should be, “If the International Code of Nomenclature of Prokaryotes (2008 Revision) [doi:10.1099/ijsem.0.000778] permits”, or similar.

We thank reviewer for the comment and suggestion. We have modified the para as mentioned as per the suggestion to “If the International Code of Nomenclature of Prokaryotes (2008 Revision) [doi:10.1099/ijsem.0.000778] permits”

Validating the deep-phylotaxonogenomic …

“While the advent of genomics era has allowed to propose taxonomic cut-offs for species and genera, robust genomic indices for higher taxa like families and above are not yet available.” Awkward. Perhaps instead, “The advent of genomics has led to quasi-standard genomics-based cut-offs for species (95% ANI) and genera (60%-80% AAI), there are no widely accepted cutoffs for higher taxa.” “These are: 94.5% for genus, 86.5% for family, 82.0% for order, 78.5% for class and 75.0% for phylum [34].” See the comments I have labeled “Major Issue 2”.

We thank reviewer for the comment and suggestion. We have modified the para as mentioned as per the suggestion to “The advent of genomics has led to quasi-standard genomics-based cut-offs for species (95% ANI) and genera (60%-80% AAI), there are no widely accepted cut-offs for higher taxa.”

Oleiagrimonas soli and Xyllela fastidosa are boundary cases in your analysis. Discuss.

We thank the reviewer for the comment and suggestion. We have now removed the Oleiagrimonas soli from the emended genus Frateuria as it is not meeting the monophyly rule and was forming outgroup in phylogenomic tree(s) and 16S rRNA tree. However, we Xylella fastidiosa was included emended genus as it is meeting the requirement of monophyly rule in phylogenomic tree(s) and 16S rRNA tree apart from meeting the 16S rRNA sequence similarity threshold. In fact, Xylella fastidiosa is clustering with Xanthomonas deep within the clade in the phylogenetic trees. We have now discussed these points in the revised manuscript (line 258).

We are not including the Oleiagrimonas soli in the emended genus Frateuria as it is not meeting the strict requirement of monophyly rule and was forming outgroup in phylogenomic tree(s) and 16S rRNA tree (Fig 1, Fig 2, Fig. 3 and Fig 6). However, we Xylella fastidiosa was included in the emended genus as it is meeting the strict requirement of monophyly rule in phylogenomic tree(s) and 16S rRNA tree apart from meeting the 16S rRNA similarity threshold. In fact, Xylella fastidiosa is clustering with Xanthomonas deep within the clade in the phylogenetic trees. We have now discussed these points in the revised manuscript.

 “In fact, the monophyly test as revealed by major sub-lineages corresponding to these genera indicate that we have been conservative in selecting the AAI and 16s rRNA cut-offs.” The is unnecessarily vague. Examining Supp table 4, I can imagine what you mean, but it either be explicit or delete the point. Also, see “Major Issue”.

We thank the reviewer for the comment. We have now deleted the statement.

 “saturation through discoveries”. I have no idea what this means. Please restate.

We thank the reviewer for the comment. We have now deleted the statement.

Figures

Figure 1.

The image is fuzzy, please provide a higher resolution image

Triangles of various size are used to represent the bootstrap values on the tree. However, the sizes of the triangles are indistinguishable. Please provide the numeric values or use colored triangles to enable the reader to discern these values.

We thank you for the comment. We have now used figure with high resolution and instead of triangles, we have now provided numeric bootstrap values (Figure 1) in revised manuscript

“blue triangles representing species in the genus Xanthomonas.”, “the red triangles represent the Rhodanobacter genus.” The caption text should be changed to indicate that the authors are proposing that these species be places in the genera Xanthomonas and Rhodanobacter.

We thank you for the comment. We have changed the legend indicating color to their respective genus.

Figure 2.

Same as Figure 1: The image is blurry. The labels are unreadable. The triangles are indistinguishable. In fact, the triangles are so small they are invisible. Move a high-res image to Supplemental?

The caption: Ditto figure 1 - the placement of species in genera is proposed.

We thank you for the comment. We have now provided figure with high resolution and instead of triangles, we have now provided numeric bootstrap values (Figure 2) in revised manuscript

Figure 3.

Ditto Figures 1 and 2.

We thank you for the comment. We have now used figure with high resolution and instead of triangles, we have now provided numeric values (Figure 3) in revised manuscript

Figure 4.

Ditto Figures 1-3. Also, the values on the heatmap scale are illegible. Furthermore, is there a good reason why the AAI and cAAI heatmaps order the strains differently?

We thank you for the comment. We have now used the high-resolution image and we are provided heat map for AAI and cAAI, with values inside.

Figure 5.

Ditto Figures 1-4. I think that it is extremely unlikely that you will be able to make this figure legible while it is in the main text. Please consider including a (very!) high-resolution image in Supplemental.

We thank you for the comment. We have now used the high-resolution image with values inside heat map

Figure 6.

Fuzzy. Please provide a sharper image.

Why was 88.07 for P. aeruginosa omitted?

We thank you for the comment. We are providing a high-resolution image (Figure 6). We omitted P. aeruginosa value in tree as we are considering it as outgroup although we have provided values in supplementary table.

Supplemental Table 3

I note the following genera that you propose to move between families:

Rhodanobacteraceae -> Lysobacteraceae

  • Aquimonas voraii DSM 16957_T
  • Chiayiivirga flava DSM 24163_T
  • Rehaibacterium terrae DSM 25897_T

Lysobacteraceae -> Rhodanobacteraceae

  • Metallibacterium scheffleri DSM 24874_T
  • Mizugakiibacter sediminis skMP5_T
  • Pseudolysobacter antarcticus strain:AQ6-296_T

I do not find discussion of any of these in the paper. Find rectify this.

We thank you for the comment. We are now providing discussion about all these genera as well in revised manuscript- ‘Re-Shuffling of six genera within family Lysobacteraceae and Rhodanobacteraceae’ (Line: 170)

Major Issue 1

“cut-off values of 65% used for delineating novel genus”.

Why/how was this cutoff chosen? Previously you reported works that use cutoffs ranging from range 60%-80%. The choice of this cutoff is one of the most important details of this paper, and there is no detailed discussion about how you arrive at a cutoff value of 65%. It would appear to be arbitrary, which significantly undermines your stated claim that the species and genera need to be reorganized the status quo.

Consider, is 65% the only cutoff less than 80% that satisfies the monophyletic requirement that you have stated? If it is, then please demonstrate this, because I believe that it would be the most convincing evidence to support your reorganization. If it is not, then shouldn’t a more conservative cutoff be selected that results in fewer changes to existing taxonomy?

We thank the reviewer for the comments and suggestion. We have now revised the manuscript to address the issue.

In the revised manuscript we discuss that cut-off value 65 % and above for AAI satisfies the monophyletic requirement for the inclusion in the emended genus Xanthomonas and emended genus Frateuria as the members are forming a cluster or sub-lineage in the phylogenomic trees (Fig 1, Fig 2 and Fig 3) and also 16S rRNA tree (Sup Fig 1). Further 65% and above AAI cut-off is also matching with the 16S rRNA threshold (Figure 6 and Supp. Table 4) for delineating genera and even in the complete 16S rRNA based tree the members of the expanded genera form a clade (Fig. 6). Now we have mentioned and discussed the same in the revised manuscript.

Major Issue 2

“These are: 94.5% for genus, 86.5% for family, 82.0% for order, 78.5% for class and 75.0% for phylum [34].”

While I accept these are the thresholds used in [34], I would claim that these thresholds, when applied using your own method, give nonsensical results. Your results (Suppl table 4) and these thresholds would place P. aeruginosa and X. campestries not only in the same order, but in the same family. You more or less state this in the discussion.

In order illustrate the problem, I used the GGDC server to compute similarity measures for the 16s sequences downloaded from LPSN for E.coli, P.aeruginosa, and X.campestries, which are all in same class but each in a different order. Here are the results:

ecoli

paeruginosa

xcampestris

ecoli

100

88.45

88.7

paeruginosa

88.45

100

88.07

xcampestris

88.7

88.07

100

Again, your method would place all of these in the same family.

A small improvement would be to use another method for computing 16s similary. Here are two possibilities:

NCBI “blastn”:

ecoli

paeruginosa

Xcampestris

xfastidiosa

ecoli

100

85.135

84.963

84.019

paeruginosa

85.135

100

85.267

84.913

xcampestris

84.963

85.267

100

95.678

xfastidiosa

84.019

84.913

95.681

100

“water” (pairwise Smith-Waterman, found in the EMBOSS package):

ecoli

paeruginosa

xcampestris

xfastidiosa

ecoli

100

84.4

83.7

82.2

paeruginosa

84.4

100

84.6

83.9

xcampestris

83.7

84.6

100

95.4

xfastidiosa

82.2

83.9

95.4

100

Either of these gives much more conservative results than GGDC, and these results + the [34] cutoffs give results much closer to the status quo. E.g., for “water”, each pair in the same…

ecoli

paeruginosa

Xcampestris

xfastidiosa

ecoli

strain

Order

Order

order

paeruginosa

order

Strain

Order

order

xcampestris

order

Order

Strain

genus

xfastidiosa

order

Order

Genus

strain

This is better, but still nonsense: E.coli, P.aeruginosa, and X.campestris are placed in the same order.

“However, the same was not the case with the families know in order Lysobacteraceae. In the absence of genomic indices for taxa above genus level, the proposed 16s rRNA boundaries question of the existence or earlier proposal of family Lysobacteraceae and Rhodanobacteraceae as the cut-off of the members are way above 86.7%[34].”

Your 16s results and discussion would lead the reader believe that you are suggestung reconsidering the entire taxonomy of class Gammaproteobacteria. I would respectfully suggest that your approach (GGDC + the cutoffs from [34]) should be reconsidered instead. Everything in this manuscript related to 16s similarity needs to be redone - methods, results, and discussion, all of it. Your results would have well-established orders of Gammaproteobacteria devolving into a single family (at worst) or order (at best), which is unreasonable. Therefore, these (unreasonable) results do not support your (more reasonable) adjustments within Lysobacterales (Xanthomonadales).

We thank the reviewer for the critical comments and valuable suggestions.  Now have reworked on the methods, results and discussion. As suggested by the reviewer, we have now used NCBI blastn and replaced with new figure and table in this regard. We also agree with the reviewer that application of the threshold is misleading at the level of family and order. Hence, we are now restricting the application of 16s threshold for genus level adjustments and not for the families or the order in the revised manuscript.

Round 2

Reviewer 1 Report

Dear authors,

Thank you for addressing the comments made in the previous version, you provide a much cleaner second version, however, I still believe there are loose ends that should be addressed.

First, how the readership can implement the methods you used in a way that allow reproducibility and consistency of the analyses? I believe you should provide more information in this regard, specially considering that in the future novel species would be described for this genus.

In your reply 6: "We thank the reviewer for the comment and suggestion. As our primary aim was to investigate genus level classifications and above. Since dDDH is not used investigate genera and above taxa but for species/sub-species delineations, we have not used in our study."

I understand the scope of this research work, however you are making such an announcement by claiming that several genera now belong to Xanthomonas without supporting your result by discussing whether there might be synonym within the genus or not. I strongly believe that the revision should go this deep.

Line 38

Reference [8-9-10] should be as in Line 43.

In the main text, please refer to the type strains with the T in upper script, even though in the Figure is written _T.

Author Response

We thank the reviewer for their valuable comments, suggestions and feedback on our manuscript towards the improvement of the manuscript. we have addressed the comments and incorporated the suggestions of reviewer. Reviewers comments are in blue and our response are in black.

Reviewer 2 Report

Thanks to the authors for their thoughtful responses to my comments. I have the following, relatively, minor suggestions that I hope will improve the current manuscript.

Line 38: "[8-9-10]" should be "[8-10]".

Line 42: strike "the ability to cause".

Line 46: "Recent research has shown that Lysobacter species, L. enzymogenes C3 have the potential to act as biological pest controllers for plant diseases [15]." I am confused - isn't C3 a strain of Stenotrophomonas maltophilia? If it has been re-identified as Lysobacter enzymogenes, please provide a reference.

Line 71: "is enabling" should be "enables"

Line 79-82: "Further AAI values for the core genes (cAAI), wherein AAI is typically calculated from all genes shared between a pair of organisms, or family, or order, or a taxon under consideration [28, 33] is being employed as a robust parameter for genus delineation."  Very awkward and grammatically incorrect. Can it be written as, "Further, whereas AAI is typically calculated from all genes shared between a pair of organisms [28, 33], AAI values for the core genes (cAAI) are being employed as a robust parameter for genus delineation." I am not sure where "or family, or order, or a taxon under consideration" would be placed.

Line 87-88: "and the availability of a robust cutoff for delineat- ing higher taxa from genus to phylum [37]". This is at odds with what is stated on lines 275-276. Please reconcile.

Line 131-132: "The 16S rRNA phylogeny and similarity were generated using BLASTn." Do you actually include a phylogeny based on 16S? If not, please strike that. Also, did you use the web or stand-alone version of BLASTn. If the later, which version? Finally, please include a reference for BLASTn.

Figure 1: The text is still blurry. The color-coded branches work nicely.

Figure 2: The text is still blurry. The colors on the branches are indiscernible. Provide a higher-resolution image that that the reader can read the text, and consider dropping the bootstrap values from this figure. As an alternative a very high-resolution figure that included the colored branches could be included in the supplemental files.

Figure 3: Ditto figure 2.

The paragraph starting at line 191: I found this paragraph very confusing. As written, I do not find it clear what current composition of Lysobacteraceae and Rhodanobacteraceae. E.g., "The family Lysobacteraceae includes [...] Rudaea [...] The family Rhodanobacteraceae includes the genus [...] Rudaea [...]". It's a member of both? I think that it would be more clear if you directly stated, "Based upon the previous results, we propose to move the genera ... from Lysobacteraceae to Rhodanobacteraceae and to move the genera ... from Rhodanobacteraceae to Lysobacteraceae. Placing the references within parentheses would also improve readability, I beleive.

Figure 5: The text and tree are so pixelated that they completely uninformative. I suggest putting a figure with a small subset of the strains in the manuscript and include a very high-res image containing all strains in the supplemental.

Line 269: "16SrRNA" should be "16S rRNA".

Line 271-272: "there are no widely accepted cut-offs for higher taxa. These are: ..." There is a transition missing here. "These" appears to refer to "cut-offs for higher taxa", but you just said that none are "widely accepted". Are you trying to say that Yarza, et al. have proposed these cutoffs but that they are not widely accepted?

Line 275-276: "However, the same is not the case with the usage of 16S rRNA thresholds for higher-level taxa, i.e., above genus." This is at odds with what is stated on lines 87-88. Please reconcile.

Line 281: "16SrRNA" should be "16S rRNA".

Line 289-290: "We have now discussed these points in the revised manuscript." Strike.

Figure 6: Ditto figure 2. Also, you state on lines 275-278, "However, the same is not the case with the usage of 16S rRNA thresholds for higher-level taxa, i.e., above genus. Hence, in the present report, we are only proposing an emended or expanded description of the genera Xanthomonas and Rhodanobacter and not proposing any family or emended description of the higher taxon." So, why do you report the 16S similarity values for the family, Xanthomonadaceae? Since there is no widely accepted cutoff for 16S at the family level, how is the reader supposed to interpret this data?

Line 301: "Xanthomonadales [Saddler and Bradbury 2005 a, b)". Mismatched "[" and ")".

Line 329-330: "The proposed genus including Frateuriaceae includes the genus Frateuria". I think this should be "The proposed family Frateuriaceae includes the genus Frateuria".

Line 364: Should this line be in a bold font? Or a new section heading?

Line 369: "The type genus is Frateuria aurantia (ex Kondô and Ameyama 1958) Swings et al. 1980". Strike this sentence as it is already stated on lines 366-367.

Line 381: "16SrRNA" should be "16S rRNA".

Author Response

We thank the reviewer for the valuable comments, suggestions and feedback towards the improvement of the manuscript. We have addressed the comments and incorporated the suggestions of reviewers, reviewers comments are in blue and our response are in black.

Round 3

Reviewer 1 Report

Dear Authors,

Thanks for addressing my comments.

With best wsihes,

Reviewer 2 Report

I wish to thank the authors for incorporating my suggestions. I hope that the authors feel, as I do, that this manuscript is much improved from the original version.

My only request is that you add the version number of PIRATE to its mention in the text.

I look forward to seeing this paper in "print" and to see the reaction that it will generate.